# The impact of environmental regulation on regional economic growth: A case study of the Yangtze River Economic Belt, China

**Jiajun He[1], Zirui Huang[1], Xin Fan[1], Hui Zhang[2]\*, Rong Zhou[3], Mingwei Song[4]**

**1** School of Management, Wuhan Institute of Technology, Wuhan, China, **2** School of Chemistry and Environmental Engineering, Wuhan Institute of Technology, Wuhan, China, **3** China International Engineering Consulting Corporation, Beijing, China, **4** College of Resources and Environment, Huazhong Agricultural University, Wuhan, China

\* zhangh73@hotmail.com

**Data Availability Statement:** All data files are available from the figshare database (https://doi.org/10.6084/m9.figshare.23702052)

**Funding:** This research was funded by the Fundamental Research Funds for the Central

## Abstract

In this paper, we take the Yangtze River Economic Belt as the study area and analyze three types of environmental regulation tools, namely, command-and-control (CAC), market-incentivized (MI) and public-type (PT). We apply the threshold effect to test the impact of each of these tools on regional economic growth and analyze the relationships between the tools and environmental regulation. The entropy method is used to calculate the comprehensive environmental pollution index of each province and city in the Yangtze River Economic Belt. Using Stata 14.0 measurement software and based on provincial data with respect to the Yangtze River Economic Belt from 2014 to 2021, a panel threshold model is used to test the impact of the three types of environmental regulation tools on regional economic growth and analyze the relationship between environmental regulation and regional economic growth. It is found that the relationship between environmental regulation and economic growth is non-linear. There is no significant relationship between CAC environmental regulation and regional economic growth; there is a single threshold effect between market-incentive environmental regulation and public participation environmental regulation on the economic growth of the Yangtze River economic belt.

## 1. Introduction

The report of the 20th National Congress of the Communist Party of China (CPC) suggested that Chinese-style modernization is a type of modernization in which people and nature live in harmony, and we should protect nature and the ecological environment, and follow a path of civilized development that includes production development, rich living and good ecology to achieve the sustainable development of the Chinese nation [1]. As of 2021, the Chinese gross domestic product was 114.37 trillion Yuan, an increase of 8.1% over the previous year. With the massive exploitation of resources, China's environmental pollution problems have become prominent, and China should follow a path of ecological environmental protection that is in line with the characteristics of China's economic growth [2].

Universities (2662022ZHYJ004) and the National Natural Science Foundation of China (40971054). The funders had no role in study design, data collection and analysis, decision to publish, or preparation of the manuscript.

**Competing interests:** The authors have declared that no competing interests exist.

Given the public goods nature of environmental resources, the negative externality of environmental problems and the existence of the opportunism of microeconomic agents, there is a phenomenon of "free-riding" in the management of environmental pollution by various economic agents [3], resulting in environmental problems not being solved. For the market alone, it is difficult to achieve the goal of pollution reduction. Environmental regulation is needed to make up for the defects of "market failure" [4]. Environmental regulations are related to China's economic development. China's economy has undergone an important period of transition from high-speed development to high-quality development, and it is not easy to develop both green measures and the economy simultaneously. Environmental regulation is related to China's economic development, and in order to continue to improve our economy [5], we must choose the appropriate environmental regulation tools, so as to maximize the effectiveness of environmental regulation.

The Yangtze River Economic Belt is a major national strategic development area in China, with a population and GDP accounting for over 40% of China's total. However, the literature on the Yangtze River Economic Belt is not yet abundant, with most of the literature focused on provinces, urban agglomerations or a certain industry in China. There are relatively few studies on specific regions. Therefore, it is important to study the relationship between environmental regulation and regional economic growth and its improvement path to promote the construction of an ecological civilization. In this paper, a comprehensive environmental pollution index is included in the study of environmental regulation and regional economic growth, and the entropy method is used to obtain the environmental pollution index to investigate the complete environmental pollution situation of provinces and cities in the Yangtze River Economic Belt from 2014 to 2021. We examine the impact of different regulatory tools on economic growth. In order to avoid bias in the estimation results caused by traditional subjective judgment of environmental regulation intervals, the threshold effect model proposed by Hanse is used to determine the threshold value based on empirical analysis, in order to obtain the optimal interval of economic growth under different environmental regulation tools.

## 2. Literature review

### 2.1. Study of environmental regulation types

Environmental regulations can be classified into three types according to the stringency of enforcement; namely, CAC environmental regulations, MI environmental regulations and PT participation environmental regulations [6]. The current research studies the implementation effects of these three environmental regulations from the perspectives of ecology, energy conservation and emission reduction. Ren et al. [7] studied the impact of these three types of environmental regulations on eco-efficiency in eastern, central and western China from the perspective of regional eco-efficiency, and finally found that the three types of environmental regulations have their own advantages, and the impact of different types of environmental regulations on eco-efficiency is different in different regions. On the basis of covering a variety of environmental regulations, this study considers the timeliness of environmental regulations, introduces a lag period of environmental regulations, and investigates the dynamic effects of environmental regulations on regional ecological efficiency. Guo et al. [8] compared the impact of CAC environmental regulations and MI environmental regulations on greenhouse gas emission reduction. He believes that stricter CAC environmental regulations and moderate MI environmental regulations are favored. The study covers a variety of environmental regulations and measures each type of environmental regulation with an environmental policy stringency (EPS) index, taking into account nonlinear effects. Hua et al. [9] analyzed the impact of three environmental regulations on sewage discharge. The results show that all three

environmental regulations have certain effects, among which public participation is the most effective. However, the effectiveness of the three environmental regulations varies in different regions. The study analyzed and compared the impact of various environmental regulations on sewage discharge, and considered the synergistic effect and regional heterogeneity of environmental regulations.

## 2.2. Research on environmental regulation and economic development in different countries

For the 11 fastest growing emerging economies (N11), such as South Korea, Mexico, and Türkiye, Eregha et al. [10] studied how environmental regulation affects the ecological footprint (EFP) in N11 countries. He found that environmental regulations are not effective in slowing down environmental degradation. Economic growth, trade and energy consumption inject vitality into the EFP. The research results confirm that energy consumption and economic growth promote environmental degradation in all N11 countries .

India is the third-largest greenhouse gas emitter in the world, and the carbon emissions generated by the consumption of a few wealthy individuals are about seven-times that of the poorest households. The Indian the government should increase environmental regulations, formulate appropriate environmental policies (green economy), develop a renewable economy through energy conservation and energy-saving technologies and reduce the use of non-renewable energy by the wealthy [11].

In New Zealand, in recent years, its economic policies have become increasingly concentrated, with local government powers shrinking, reflecting a new regional development mindset of attracting foreign investment based on the market. Policies have become more centralized and market-oriented. At the national and local levels, economic development and environmental policies tend to be market-oriented and involve enterprise-focused support [12].

Different countries and regions have different attitudes towards the use of environmental regulations. The impact of environmental regulations on economic development is also different.

## 2.3. Study on the relationship between environmental regulation and economic growth

Domestic and foreign research on the impact of environmental regulation on economic growth can be divided into three types.

The first is the "cost-following theory": Environmental regulation raises the production cost of enterprises and is not conducive to improving business performance. Xie et al. [13] found that the relationship between environmental regulation and economic growth in China is in accordance with the spatial perspective of the cost-following theory through the spatial expansion of environmental regulation. The spatial effect of environmental regulation is very important for the study, which breaks through the spatial constraints of previous studies and is helpful for the formulation of environmental regulation policies in neighboring regions. The study also contributes to the issue of reconciling environmental regulation and economic growth in neighboring regions. Zhu and Zhao [14] analyzed the influence of environmental regulations on financial performance and found that environmental regulations inhibited the financial performance of enterprises, creating a negative effect of cost extrusion, while technological innovation could partially offset the "cost squeezing effect" and mitigate the inhibitory relationship of environmental regulations on financial performance. This study added technological innovation elements into the relationship between environmental regulation and

economic growth, and pointed out a feasible way for enterprises to balance the relationship between environmental regulation and economic growth.

The second is the "innovation compensation theory": Environmental regulation stimulates firms to engage in technological innovation to reduce the scope of pollution and control their production costs. Chen et al. [15] studied the influence of environmental regulation on enterprises' green technology innovation. This paper considers the impact of environmental uncertainty, and the research results show that environmental uncertainty has a positive impact on enterprises' green technology innovation. Due to the complexity of China's region and the change of environmental policies, this study has a strong guiding role for enterprises that insist on technological innovation in China. Jiang et al. [16] analyzed the impact of environmental regulation on green technology innovation using data in respect of 30 provincial administrative panels in China from 2008 to 2017. The results show that there is a U-shaped relationship between environmental regulation and green technology innovation. The strengthening of environmental administrative decentralization and environmental supervision decentralization will enhance the promoting role of environmental regulation on green technology innovation. The study concluded with an important reference value for determining reasonable levels of environmental decentralization among different regions and improving relevant environmental regulation strategies. From the perspective of China's agricultural green productivity, Peng [17] analyzed the impact of environmental regulation on the growth of China's agricultural green productivity and its mechanism. He found that environmental regulation significantly promoted the growth of agricultural green productivity by promoting the progress and innovation of agricultural green technology. This paper focuses on the regulatory effect of agricultural innovation investment on environmental regulation of China's agricultural green productivity growth. Because China is a large agricultural country, this study is of great significance for China to rely on environmental regulation to promote agricultural green productivity growth.

Thirdly, the "pollution paradise hypothesis" means that enterprises in pollution-intensive industries tend to be established in countries or regions with relatively low environmental standards. Zheng et al. [18] tested the regulating effect of environmental regulation on foreign direct investment (FDI) and green economy efficiency using the spatial Durbin model (SDM) and found that environmental regulation has a positive conditioning effect on the whole. It enhanced the pollution paradise effect of FDI on the efficiency of the green economy. This study also discusses three kinds of environmental regulations on economic growth. Due to the current market background of encouraging foreign investment in China, this study contributes to the development of environmental regulations for the current situation.

In addition, scholars have found a non-linear relationship and regional heterogeneity between environmental regulations and economic growth. Liu et al. [19] explored the impact of environmental regulation on high-quality economic development. He found that environmental regulation has an obvious effect on high-quality economic development in central and eastern regions, while this phenomenon is not significant in western regions. The same type of moderating variables show obvious heterogeneity among different regions. Environmental regulation has significant regional heterogeneity in respect of economic growth.. Xie et al. [20] evaluated the "green" growth rate and productivity of industries in 30 provinces in China. He believes that China's development has significant regional heterogeneity, and that MI environmental regulations promote a much stronger productivity effect than CAC environmental regulations. Reasonable environmental regulations can enhance the competitiveness of enterprises. The study considered a comprehensive perspective, not only found regional heterogeneity, but also discussed the impact of different environmental regulations on green productivity. Liu et al. [21] studied the relationship between green finance, environmental

regulation and high-quality economic development, and found that environmental regulation plays a nonlinear regulatory role in the relationship between green finance and high-quality economic development, with a single threshold. Wang et al. [22] explores the role of environmental regulation in the widening of the North-South economic gap in China and finds that the regulation of environmental regulation has a significant convenience in narrowing the North-South economic gap. In addition, he found a positive U-shaped nonlinear relationship between the two rings. Different from the above study, this study discusses the problem of environmental regulation and economic development from the premise of regional heterogeneity, and finds the nonlinear relationship between the two.

## 2.4. Conclusion

It can be seen from the above literature that most current studies consider the relationship between a single environmental regulation and economic growth, while few studies involve the relationship between multiple environmental regulations and economic growth, and most of these studies are conducted from a national perspective China has a vast territory and a complex economic development situation. Different strategic development regions have different attitudes and sensitivities to environmental regulation. In the study of environmental regulation and economic development, the number and space of samples selected are too large, the regional heterogeneity will be further amplified in the study and the conclusions may lack pertinence and accuracy. The Yangtze River Economic belt, as a major national strategic development region, is one of the "three strategies" implemented by the central government. The relationship between environmental regulation and economic development lacks specialized study.

In summary, there are few literatures on the relationship between different types of environmental regulation and regional economic growth , which is the key to environmental policymaking [23]. For this reason, this paper covers three different types of environmental regulation instruments: CAC, market-incentive and public participation. We focus on the Yangtze River Economic Belt, an important economic development region. We attempt to analyze the impact of different types of environmental regulation on economic growth.

# 3. Materials and methods

## 3.1. Research hypothesis

In order to clarify the relationship between environmental regulation and economic growth, Xiong [24] constructed an environmental regulation intensity index and analyzed the relationship between environmental regulation and economic growth. However, the impact mechanism of environmental regulation on economic growth was not considered. An a priori assumption was made that there is a non-linear relationship between environmental regulation and economic growth, lacking theoretical basis. Yuan and Liu [25] explored the impact of cost-based and investment-based environmental regulations on economic growth, observing that cost-based environmental regulations have no significant impact on economic growth, while investment-based environmental regulations significantly promote economic growth. Enterprises are the main body of regional economies, and as they face increasingly strong environmental regulations, how to respond to the challenges brought by environmental regulations is the key issue they need to solve. Due to the fact that technological innovation can solve environmental problems at the minimum cost while also improving the core competitiveness of enterprises, technological innovation has become a strategic choice for enterprises to achieve "win-win" environmental and economic performance. Against the backdrop of increasingly severe environmental pollution, guiding enterprises to vigorously carry out green

technology innovation activities plays an important role in promoting economic growth and social progress. After reviewing the literature, it was found that although there have been a large number of empirical studies on the impact of environmental regulations on economic growth, there is little literature on the role of enterprise technological innovation in economic growth, and there is less discussion on the impact of different types of environmental regulations on economic growth. In this article we draw inspiration from the research results of You and divide environmental regulation into three types: CAC environmental regulation, MI environmental regulation and PT environmental regulation. The logical relationship between them and economic growth is established separately.

After the introduction of command-based environmental regulation standards, enterprises need to invest more in the pollution control budget in order to meet existing standards, cutting some of the technological research-and-development costs that are crucial to the long-term development of enterprises but have no obvious short-term benefits. To a certain extent, this inhibits the iteration and upgrading of enterprise products and affects enterprise income, which is not conducive to enterprise economic growth. When command-based environmental regulation is implemented, the cost of environmental pollution control decreases year by year, and the investment in production and technological innovation increases year by year, and when both inputs reach a stable critical point, the overall level of regional economic development remains stable and improves [26]. Based on this, the first hypothesis is proposed.

**Hypothesis 1:** The implementation of CAC environmental regulations has a suppressive effect on economic growth; the suppressive effect is significant in the early stage but decreases after reaching a certain critical point.

The most representative market-based environmental regulation tool is the "emission fee". This is the emission right acquired by enterprises through the purchase of pollution emission rights; it increases the production cost of enterprises to a certain extent and leads to insufficient capital investment for technological innovation, thus hindering the growth of enterprise output. After a period of payment of emission fees, the cost of enterprises for environmental management tends to stabilize and reaches a critical point, and enterprises compete to imitate or learn the production technology of advanced enterprises in order to gain competitive advantage. This promotes the economic growth and structural optimization of the industry, and the inhibitory effect of environmental regulation on the economic growth of the industry diminishes. Based on this, we propose the second hypothesis.

**Hypothesis 2:** There is an inhibitory effect of the implementation of MI environmental regulations on economic growth; the inhibitory effect is significant in the early stage but weakens when it reaches a certain critical point.

Since pollution has negative externality [27, 28], enterprises' pollution behavior causes inconvenience to public life. The public will choose to inform the relevant government departments by means of complaints and letters, and the government will propose corrective measures or penalties for enterprises creating excessive pollution after verification [29], which will increase the pressure on enterprises' production costs to a certain extent and reduce their profitability. After the implementation of public-participation-based environmental regulation, enterprises face the dual situation of stabilizing pollution control inputs and increasing corporate profits, forcing them to improve their original production methods and increase production efficiency, which encourages the regional economic level to remain stable and improve [30]. Based on this, we propose the third hypothesis.

**Hypothesis 3:** There is an inhibitory effect of the implementation of PT environmental regulations on economic growth; the inhibitory effect is significant in the early stage, but it decreases after reaching a certain critical point.

### 3.2. Variable selection

**3.2.1. Explained variables.** Economic growth Y was used as the explanatory variable, In most domestic and foreign scholars' research, GDP, GDP growth rate or per capita GDP are usually used as important indicators reflecting economic growth. In order to take into account the rationality and measurability of the study on the impact of environmental regulations on economic growth, this paper adopts the connotation of economic growth proposed by Yang et al. [31]: Economic growth refers to the growth of a country's total service and product production within a certain time span, that is, the per capita GDP or GDP increase. Therefore, 11 provinces and cities in the Yangtze River Economic Belt were selected to measure economic growth in terms of GDP per capita in the calendar years 2014–2021.

**3.2.2. Core explanatory variables.** The core explanatory variable was environmental regulation, which was divided into CAC type (X1), market incentive type (X2) and public participation type (X3) (Table 1).

CAC environmental regulation (X1) comprises the direct regulation and intervention of enterprises' emission behavior via legislation or administrative departments according to corresponding laws and regulations. Based on the statistical caliber of the National Bureau of Statistics, the proportion of "three simultaneous" project investments in the GDP of various provinces and cities in the Yangtze River Economic Belt was selected for measurement.

MI environmental regulation (X2) aims to provide various market signals through the "invisible hand" to encourage enterprises to benefit from the implementation of technological innovation. As this is the most important means of environmental regulation in China, domestic and foreign scholars tend to use it as a measure of environmental regulation. We used the proportion of pollution charges in the GDP of provinces and cities in the Yangtze River Economic Belt to express X2.

Public environmental regulation (X3) is the act of public disclosure and litigation to prevent damage to public environmental rights. Based on the availability of data, we used the proportion of environmental letters and petitions from various provinces and cities in the Yangtze River Economic Belt to the total population of the region to measure X3.

**Table 1. Explanatory notes for key variables.**

| Variable Name | Variable Meaning | Measurement |
|---|---|---|
| Y | GDP per capita | The GDP of each province and the proportion of the regional population are logarithmic. |
| X1 | "Three simultaneous" environmental investments | (Each province and district "three simultaneous" project environmental protection investment / regional GDP)×100 |
| X2 | Sewage charges | (Emission fee revenue/regional GDP by province and region) × 100 |
| X3 | Total number of letters in relation to environmental pollution by province | (Total number of environmental letters and visits from provinces/regions/total population) × 100 |
| Z1 | Fixed-asset investment | The logarithm of the fixed asset investment of each province. |
| Z2 | External trade dependence | Provinces' (total imports + total exports)/regional GDP×100 |
| Z3 | R&D investment | (R&D expenditure/regional GDP of each province) × 100 |
| Z4 | Level of human capital (average years of schooling) | The average number of years of education received by the labor force = (the proportion of the employed population who are illiterate or semi-literate × 1.5) + (the proportion of the employed population who received primary education × 7.5) + (the proportion of the employed population who received junior high school education × 10.5) + (the proportion of the employed population who received senior high school education × 13.5) + (the proportion of the employed population who received college and above × 17). |
| Z5 | Physical capital | Physical capital per capita is taken as the logarithm and physical capital per capita is calculated as follows: $K_{i,t} = K_{i,t-1}(1-\delta_{i,t})+I_{i,t}$ |

**3.2.3. Control variables.**   In this study, five control variables related to economic growth were selected (Table 1).

Fixed asset investment (Z1): In modern economic cycle theory, investment fluctuation has always been regarded as the main cause of economic fluctuation. The rapid growth of investment affects the level of the local economy. Through the construction and purchase of fixed assets, the national economy continues to adopt advanced technology and equipment, establish new sectors, and further adjust the economic structure and regional distribution of productive forces. It can reflect the economic strength and the material and cultural life of the people, and is an important indicator to test economic growth. We adopted the logarithm of the fixed asset investment of each province and city.

Dependence on foreign trade (Z2): This index reflects the influence of international trade on the change in the environmental level. The "pollution paradise hypothesis" holds that the increase of foreign trade dependence may promote the development of pollution-intensive industries and increase the environmental pollution in the region, thus affecting the economic growth of the region. Therefore, the dependence on foreign trade was included in the control variable, and the proportion of total import and export to regional GDP was measured and logarithmically processed. The total imports and exports were converted using the exchange rate between the Chinese yuan and the US dollar in the current year.

R&D funding (Z3): Modern economic growth theory shows that technological progress and knowledge accumulation are important factors in determining economic growth, and research and development (R&D) is the main source of technology and knowledge. We measured the ratio of R&D expenditure to the regional GDP and performed logarithmic processing.

Human capital level (Z4): With the changes of social structure and market supply and demand, the original mainstream of labor force can no longer meet the needs of improving the quality of contemporary social development. At this time, the improvement of the quality of human capital becomes the new power source and endogenous growth power of economic growth in the new era [32]. Therefore, this paper includes it in the control variable. In order to make the data of the same indicator comparable, the average education level of the labor force was divided into illiterate and semi-literate, elementary school, junior high school, senior high school and college and above, and the average years of education were set as 1.5, 6, 3, 3 and 3.5 years, respectively.

Physical capital (Z5): In order to maintain rapid economic growth, special attention must be paid to capital input. Physical capital is an asset that creates wealth and income, and it can create net profit for economic growth [33]. This is expressed as the logarithm of physical capital per capita. According to the method of Zhang [34], the stock of physical capital for the period 2014–2021 was measured for each region of China with 2014 as the base period, and the formula is as follows:

$$K_{i,t} = K_{i,t-1}(1 - \delta_{i,t}) + I_{i,t} \tag{1}$$

where A is the investment in region i in year t, expressed as total fixed asset formation, and taken as the fixed asset depreciation rate of 9.6%; B is the fixed asset depreciation rate in region i in year t; C is the capital stock in region i in year t; and D is the capital stock in province i in period t-1.

## 3.3. Data source

The research period of this paper was 2014–2021 and 11 provinces (municipalities) in the Yangtze River Economic Belt were taken as the sample. Data were obtained from the China

Statistical Yearbook, China Statistical Yearbook on Environment, China Environmental Yearbook, China Social Statistical Yearbook, China Population and Employment Statistical Yearbook, and the Statistical Yearbooks of 11 provinces (municipalities). For missing data, we estimated values according to the proportion.

## 3.4. Model selection

The existing literature mainly adopts three methods to study the nonlinear relationships between variables. Firstly, introducing virtual variables into the model; secondly, introducing quadratic or cubic terms of explanatory variables into the model; thirdly, using group regression analysis. However, the first and second methods are insufficient; if the quadratic, cubic and cross-multiplicative terms of the explanatory variables are introduced, this may lead to serious collinearity problems, and it is difficult to ensure the scientificity of the results. The group regression analysis method, also known as the threshold model, can effectively overcome the above shortcomings and be used to conduct empirical analysis on the interrelationships between related core variables within different intervals. Based on the above analysis, we compared several methods currently used by scholars when analyzing the nonlinear relationship between environmental regulation and economic growth and believe that using a panel threshold regression model can eliminate some shortcomings; this method is more scientific.

## 3.5. Research model design

The panel threshold model examines whether the correlation between economic growth and environmental regulation is subject to sudden structural changes. The estimated coefficients of environmental regulation and economic growth may change significantly if the values of the examined threshold variables are higher or lower than the thresholds. Since we mainly examined the impact of environmental regulation on economic growth, drawing on the empirical ideas of existing studies on environmental regulation and economic growth, the threshold estimation model used in this paper is as follows:

$$Y_{i,t} = c_{1i} + \alpha_1 X1_{i,t} I_{i,t}(th \leq \gamma_1) + \alpha_2 X1_{i,t} I_{i,t}(\gamma_1 < th \leq \gamma_2) + \ldots + \alpha_n X1_{i,t} I_{i,t}(th > \gamma_2) + \sum_{i=1}^{5} \zeta_i Control_{i,t} + \mu_{i,t} \tag{2}$$

$$Y_{i,t} = c_{2i} + \beta_1 X2_{i,t} I_{i,t}(th \leq \lambda_1) + \beta_2 X2_{i,t} I_{i,t}(\lambda_1 < th \leq \lambda_2) + \ldots + \beta_n X2_{i,t} I_{i,t}(th > \lambda_2) + \sum_{i=1}^{5} \zeta_i Control_{i,t} + \varepsilon_{i,t} \tag{3}$$

$$Y_{i,t} = c_{3i} + \varphi_1 X3_{i,t} I_{i,t}(th \leq k_1) + \varphi_2 X3_{i,t} I_{i,t}(k_1 < th \leq k_2) + \ldots + \varphi_n X3_{i,t} I_{i,t}(th > k_2) + \sum_{i=1}^{5} \zeta_i Control_{i,t} + \theta_{i,t} \tag{4}$$

$th$ denotes the threshold variable, which is the environmental regulation variable in this paper. X1 denotes CAC environmental regulation; X2 is MI environmental regulation; and X3 is public-participation environmental regulation. $\gamma$、 $\lambda$ and $k$ are the thresholds to be estimated for a specific level of environmental regulation. $I(\bullet)$ is the indicator function. After comparing the differences in environmental regulation coefficients under different types of environmental regulation and obtaining the corresponding estimated coefficients, it is further necessary to

test the estimation of the significance of the threshold effect and the confidence interval of the threshold value. Thus, the existence of hypotheses H1, H2 and H3 can be judged.

$i$ denotes province, t denotes year, $Y_{i,t}$ denotes the economic growth of a region in a given year, X denotes different types of environmental regulations, and $Control_{i,t}$ denotes a set of control variables, where Z1 is fixed asset investment, Z2 is trade dependence, Z3 is R&D funding input intensity, Z4 is human capital level and Z5 is physical capital; $\alpha$、 $\beta$、 $\varphi$ and $\zeta$ are parameters to be estimated in the model; $u_{i,t}$、 $\varepsilon_{i,t}$ and $\theta_{i,t}$ denote random error terms of the model.

# 4. Results

## 4.1. Correlation analysis

In order to test the reasonableness of the variable selection, Pearson correlation analysis was conducted for each indicator using SPSS25 software, and the results are shown in Table 2. Regarding the correlation of variables, there is a more significant negative correlation between CAC environmental regulation, MI environmental regulation, public participation environmental regulation and economic growth, which is more in line with the expectations of this paper.

In addition, the multicollinearity among the explanatory variables was tested using SPSS25 software, and the variance inflation factor (VIF) was found to be below the critical value of 10, indicating that the model does not have serious multicollinearity (Table 3).

## 4.2. Unit root test

In this study, Fisher-PP and Hadri were selected to perform unit root tests on the variables one by one [35]. The results of the tests, given in Table 4, show that all variables passed the tests under both models, indicating that the selected variables have the characteristics of smoothness.

## 4.3. Panel cointegration

The Kao test was used for the analysis [36], and the test results are shown in Table 5. The p-values under all three models passed the significance test, indicating the existence of a long-term cointegration relationship between the variables.

**Table 2. Pearson correlation coefficient matrix among variables.**

| Variable | Y | X1 | X2 | X3 | Z1 | Z2 | Z3 | Z4 | Z5 |
|---|---|---|---|---|---|---|---|---|---|
| Y | 1.000 | | | | | | | | |
| X1 | -0.160* | 1.000 | | | | | | | |
| X2 | -0.572*** | 0.128* | 1.000 | | | | | | |
| X3 | -0.243** | 0.086 | -0.143 | 1.000 | | | | | |
| Z1 | -0.256** | 0.187* | 0.286*** | -0.013 | 1.000 | | | | |
| Z2 | 0.453*** | 0.254* | -0.274*** | 0.381*** | -0.754*** | 1.000 | | | |
| Z3 | 0.755*** | -0.423** | -0.601*** | 0.419*** | -0.621*** | 0.738*** | 1.000 | | |
| Z4 | 0.622*** | -0.165** | -0.528*** | 0.218** | -0.710*** | 0.708*** | 0.834*** | 1.000 | |
| Z5 | 0.982*** | -0.123* | -0.621*** | 0.249** | -0.214*** | 0.429*** | 0.709*** | 0.572*** | 1.000 |

**Table 3. Multicollinearity test results.**

| Variable | Indicator Description | VIF | R² | Adjusted R² | F |
|---|---|---|---|---|---|
| X1 | "Three simultaneous" environmental investments | 1.142 | 0.940 | 0.934 | F(8,79) = 155.494, p = 0.000 |
| X2 | Sewage charges | 1.919 | | | |
| X3 | Total number of letters in respect of environmental pollution by province | 2.049 | | | |
| Z1 | Fixed-asset investment | 4.315 | | | |
| Z2 | External trade dependence | 5.349 | | | |
| Z3 | R&D investment | 6.231 | | | |
| Z4 | Level of human capital (average years of schooling) | 4.387 | | | |
| Z5 | Physical capital | 2.778 | | | |

## 4.4. Threshold effect

Based on Hansen's grid search method [37], the optimal estimate of the threshold $\zeta$ was found based on the principle of the least sum of squares of residuals at a grid level of 0.0025, and the results are shown in Table 6. These show that (1) there is no significant correlation between CAC environmental regulation and regional economic growth and (2) MI environmental regulation (X2) and public participation environmental regulation (X3) pass the single threshold test at a 1% significance level; the corresponding self-sampling p-values are 0.003 and 0.01, respectively, while the double threshold effect is not significant.

Because CAC environmental regulation (X1) does not have a threshold effect, in order to judge the validity of the threshold values of MI environmental regulation (X2) and public participation environmental regulation (X3), we drew on the self-help sampling method (bootstrap) and made judgments by calculating the likelihood ratio statistic (LR), using Stata 14.0 to draw the images of the likelihood ratio function LR and the horizontal line. The 95% confidence interval of the threshold estimate is the interval constituted by the critical value of 7.35 (corresponding to the red horizontal line in each figure) at the likelihood ratio statistic LR less than 5% significance level. The estimated threshold value is valid when it lies in this interval, while the LR value lies below the critical value.

As can be seen from Figs 1 and 2, the single thresholds for MI environmental regulation and public-participation-based environmental regulation are significant at the 5% confidence level with threshold values of 0.00014 and 0.0017, respectively, and the LR indicators for each threshold value are below the critical value of 7.55, as shown by the truthfulness test of the threshold variables.

**Table 4. Variable unit root test.**

| | Fisher-PP | Hadri | Deternation |
|---|---|---|---|
| Y | 51.3141*** | 4.2069*** | / |
| X1 | 58.3141*** | 2.6923*** | / |
| X2 | 44.1504*** | 3.6788*** | / |
| X3 | 54.4630*** | 3.2469*** | / |
| Z1 | 62.0062*** | 3.5781*** | / |
| Z2 | 55.4934*** | 3.5809*** | / |
| Z3 | 49.6568*** | 3.8784*** | / |
| Z4 | 48.2844*** | 3.9494*** | / |
| Z5 | 43.4257*** | 4.0712*** | / |

**Table 5. Panel cointegration test results.**

|  | Model | t-Statistic | P-value |
|---|---|---|---|
| Kao test | Modified Dickey–Fuller t | 2.7864 | 0.0027 |
|  | Dickey–Fuller t | 3.6820 | 0.0001 |
|  | Augmented Dickey–Fuller t | 1.8625 | 0.0313 |

Because CAC environmental regulation (X1) does not have a threshold effect, the threshold regression results for market-incentive environmental regulation (X2) and economic growth and public participation environmental regulation (X3) and economic growth were tested separately, as shown in Table 7.

The results of Model-2 show that the effect of MI environmental regulation on economic growth is significantly negative at the 1% level, indicating that MI environmental regulation does not lead to the predicted economic growth effect [38]. In the interval X2<0.00014, MI environmental regulation increases the cost of pollution treatment for firms, which has a negative impact on technological innovation and firm performance. However, when X2≥0.00014, enterprises will take the initiative to carry out green technology innovation in order to make themselves stand out from the competition, and the positive effect of "innovation compensation" can partially offset the negative effect of "compliance cost", thus verifying that Hypothesis 2 holds. That is, the implementation of MI environmental regulations has a suppressive effect on economic growth; the suppressive effect is significant in the early stage, but decreases when a critical point is reached. The results of Model-3 show that the effect of public-participation-based environmental regulation on economic growth is significantly negative at the 1% level, with a single threshold. At X3<0.0017, public information asymmetry about environmental quality, if it exists, is bound to make the emission companies lose their initiative and dominance, thus affecting business performance. When X3≥0.0017, enterprises face the double pressure of reducing environmental pollution and improving enterprise performance and will improve their original production methods to reduce pollution and improve production efficiency. The cost of environmental management tends to stabilize and reaches a critical point, and enterprises will put more resources into environmental technology research and development and other aspects to improve the quality of services and meet the green requirements,

**Table 6. Threshold test results, threshold estimates and confidence interval.**

| Variable | Threshold Variable | Estimate of Threshold | P value | F value | 95% Confidence Interval | critical value | | |
|---|---|---|---|---|---|---|---|---|
|  |  |  |  |  |  | 1% | 5% | 10% |
| X1 | Single-threshold | 0.0030 | 0.86 | 3.86 | [-0.023,0.005] | 21.797 | 17.423 | 13.881 |
|  | Double-threshold | 0.0036 | 0.37 | 7.66 | [-0.010,0.007] | 29.802 | 20.872 | 15.538 |
|  | Three-threshold | 0.0029 | 0.40 | 7.38 | [-0.031,0.002] | 26.166 | 16.000 | 13.649 |
| X2 | Single-threshold | 0.00014*** | 0.00 | 21.80 | [-1.298,0.274] | 30.142 | 21.993 | 18.350 |
|  | Double-threshold | 0.0001 | 0.15 | 13.14 | [-2.726,0.212] | 24.737 | 18.762 | 16.139 |
|  | Three-threshold | 0.0002 | 0.76 | 3.25 | [-2.637,0.193] | 37.648 | 28.653 | 18.491 |
| X3 | Single-threshold | 0.0017*** | 0.01 | 36.89 | [-0.034,0.097] | 37.893 | 26.704 | 20.391 |
|  | Double-threshold | 0.0012 | 0.58 | 6.87 | [-0.083,0.095] | 27.293 | 20.086 | 17.441 |
|  | Three-threshold | 0.0015 | 0.55 | 4.39 | [-0.083,0.097] | 17.906 | 12.717 | 10.471 |

Note: both the p value and the critical value were obtained using repeated sampling with bootstrapping 300 times.

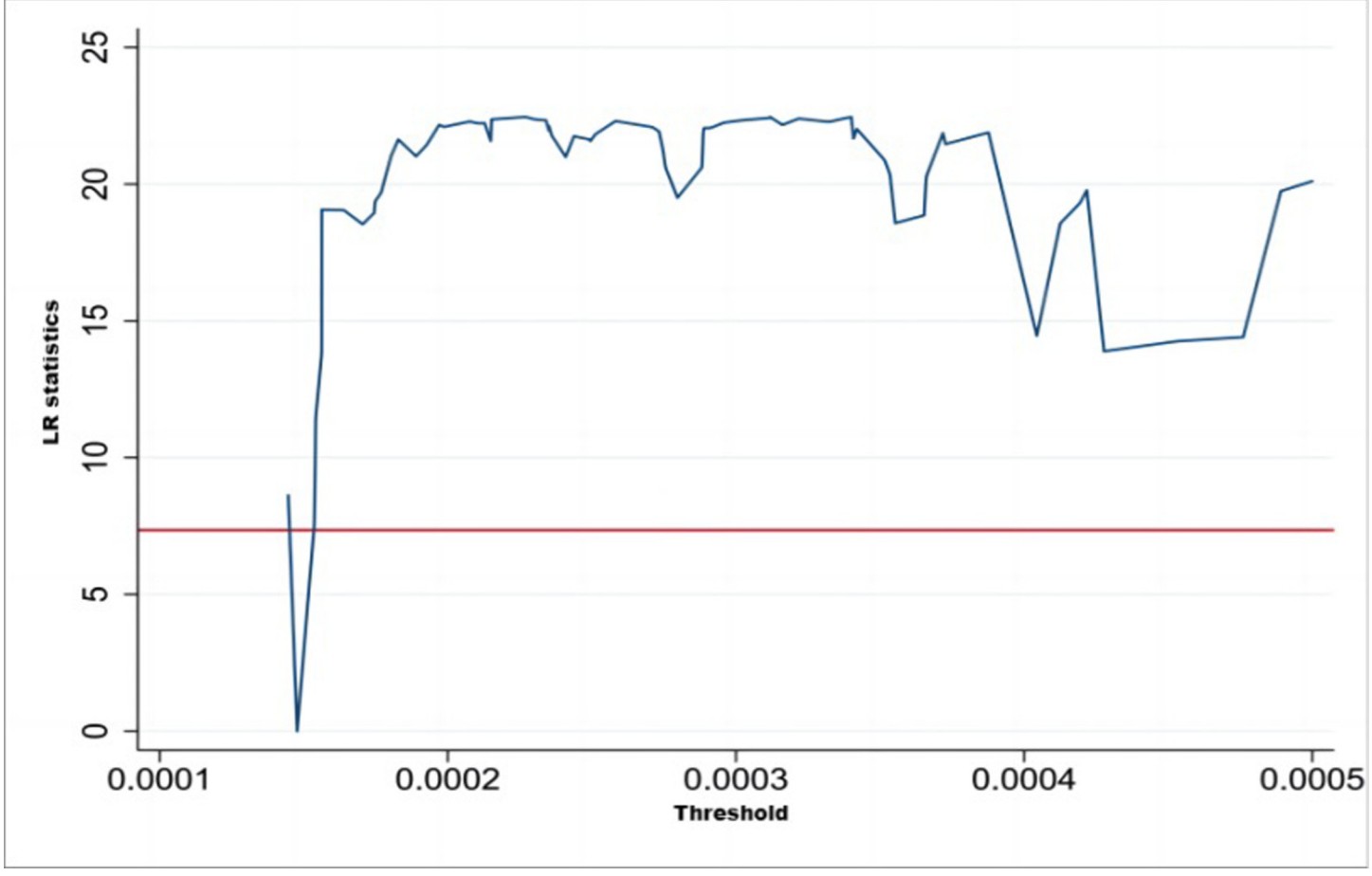

**Fig 1. Single-threshold LR indicator for market incentive environmental regulation (X2).**

which are important for sustainable regional economic growth, thus verifying that Hypothesis 3 holds. That is, there is an inhibitory effect of the implementation of PT environmental regulation on economic growth; the inhibitory effect is significant in the early stage, but weakens when it reaches a certain critical point.

The reason why this hypothesis was not tested is that CAC environmental regulations are typically characterized by the "external constraint" of administrative orders, and polluters have little choice and are forced to comply with the rules and regulations mechanically. It is also possible that the hypothesis of CAC environmental regulation and economic growth was not tested because of the regional limitations of the data for each province and city in the Yangtze River Economic Belt and the insufficient length of time of the selected data.

### 4.5. Robustness test

Based on the consideration of the robustness of the regression results, we added control variables for robustness testing by drawing on the method of Cai and Ru [39]. We introduced the control variable of the urbanization level (Z6) and used the proportion of the urban population to the total regional population to measure the urbanization level. The results of the robustness tests are shown in Table 8.

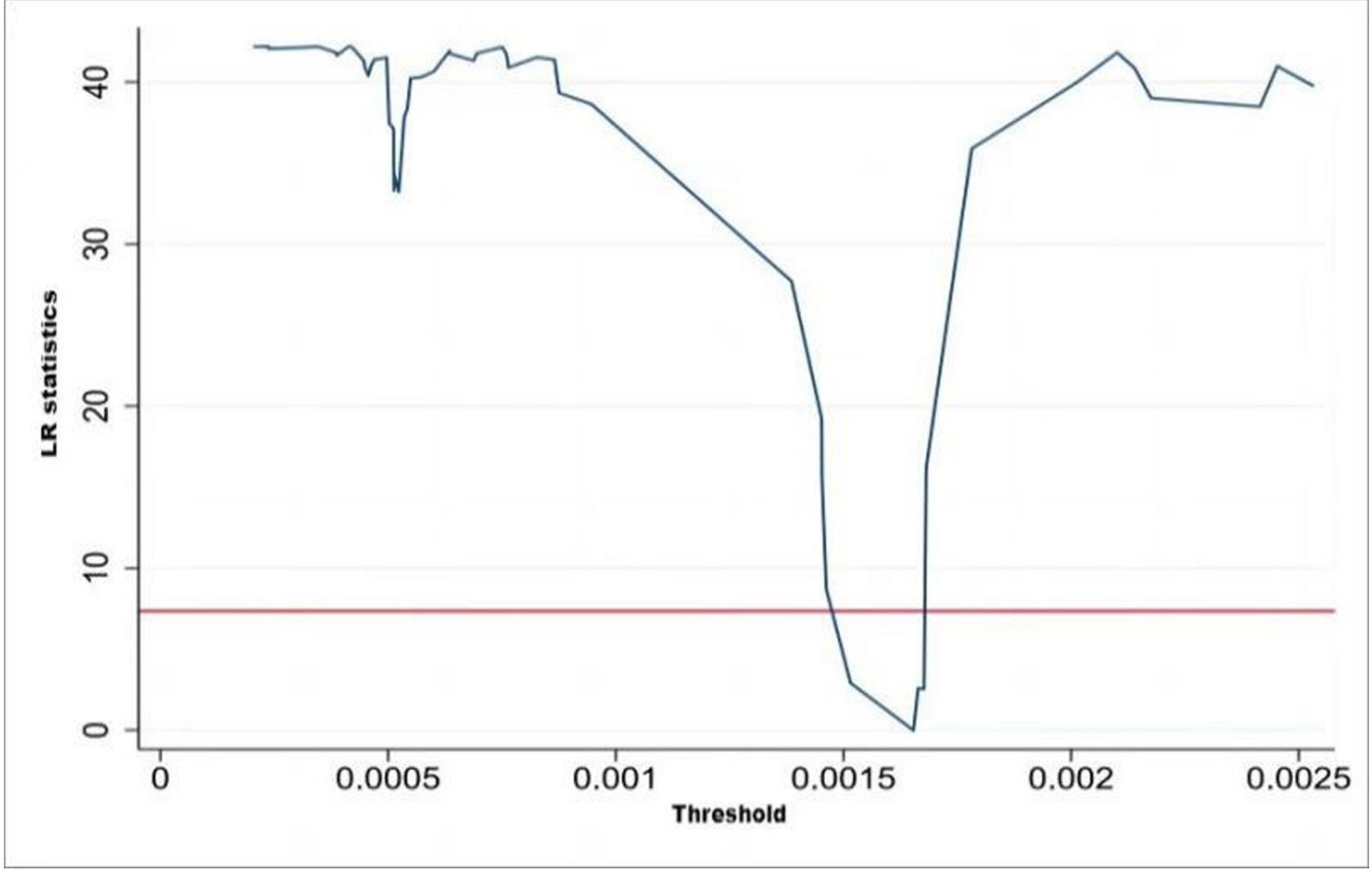

**Fig 2. Single-threshold LR indicator for public participatory environmental regulation (X3).**

Robustness tests conducted by adding control variables revealed that both Model-2 and Model-3 passed the 1% significance level, which is generally consistent with the findings above. The estimated coefficients after the robustness test differed slightly in magnitude from those of the original threshold model, but none of the directions of influence changed fundamentally, indicating the robustness of the study results [40].

## 5. Discussion

### 5.1. Impact of CAC environmental regulation on regional economic growth

We failed to verify that the hypothesis was established; the role of CAC environmental regulation on economic growth was not obvious, and we did not detect that it had a threshold effect.

CAC environmental regulation hinders the growth of enterprise total factor productivity, and this negative effect is lagging and persistent. It is difficult to achieve a win-win situation between environmentally sustainable development and enterprise total factor productivity growth under CAC environmental regulation [41], which corresponds to the conclusion that "CAC environmental regulation has no obvious effect on economic growth". Due to the lack of flexibility of CAC environmental regulation, polluters have little choice. Although adopting CAC environmental regulation can effectively improve energy conservation and emission

**Table 7. Regression testing of environmental regulation and economic growth threshold.**

| Title 1 | Model-2 (Market) | Model-3 (Public) |
|---|---|---|
| | y | y |
| Z1 | -0.149***(0.049) | -0.121**(0.048) |
| Z2 | 0.013(0.014) | 0.024(0.014) |
| Z3 | -0.132**(0.054) | -0.195***(0.054) |
| Z4 | 1.055***(0.161) | 1.028***(0.165) |
| Z5 | 0.869***(0.039) | 0.858***(0.036) |
| X2<0.00014 | -805.716***(188.499) | |
| X2≥0.00014 | -146.279**(61.962) | |
| X3<0.0017 | | -52.896***(11.686) |
| X3≥0.0017 | | -19.171**(29.435) |
| _cons | 1.684***(0.424) | -0.848**(0.381) |
| Observations | 88 | 88 |
| R-squared | 0.988 | 0.987 |

Note:

***denotes p<0.01

** denotes p<0.05

* denotes p<0.1, and the values in brackets are standard errors.

reduction, it is not conducive to the long-term development and economic growth of enterprises [42].

The negative effect of CAC environmental regulation on carbon productivity is greater than the positive effect, while MI environmental regulation has a higher positive effect on carbon productivity [43]. It can be seen that the use of CAC environmental regulation alone not only has no obvious effect on economic growth, but it also has some negative effects. Nie et al.

**Table 8. Robustness test results.**

| Title 1 | Model-2 (Market) | Model-3 (Public) |
|---|---|---|
| | y | y |
| Z1 | -0.064*(0.208)* | -0.121**(0.048) |
| Z2 | 0.018(0.013) | 0.024(0.014) |
| Z3 | -0.147***(0.051) | -0.195***(0.054) |
| Z4 | 1.424***(0.157) | 1.028***(0.165) |
| Z5 | 0.764***(0.061) | 0.858***(0.036) |
| Z6 | -0.064*(0.054) | -0.084*(0.041) |
| X2<0.00014 | -225.562***(75.612) | |
| X2≥0.00014 | -32.778**(60.482) | |
| X3<0.0017 | | -119.171***(29.435) |
| X3≥0.0017 | | -32.896***(11.686) |
| _cons | -1.044*(0.571) | -0.848**(0.381) |
| Observations | 88 | 88 |
| R-squared | 0.989 | 0.987 |

Note:

***denotes p<0.01

** denotes p<0.05

* denotes p<0.1, and the values in brackets are standard errors.

[44] believes that voluntary environmental regulation plays an important role in traditional environmental regulation. It can take on the task of improving the flexibility, autonomy and effectiveness of environmental regulation, and has a significant positive impact on the green technology innovation of enterprises. The process from passive to active needs the help of MI environmental regulation and a PT environment. The incentives of market policies and the joint efforts of the public, enterprises and other market entities can play a positive role in regional economic growth.

Some current studies have some similarities with the results of this paper. Tian et al. [45] believed that the regulation and control of environmental regulations has a "U-shaped" effect on economic growth, which first inhibits and then promotes economic growth. [46] believes that environmental regulation and economic development have regional heterogeneity, and the Yangtze River Economic Belt is mainly located in the central and southern regions of China. This paper holds that in the central region, there is an "innovation compensation" effect between environmental regulation and high-quality economic development, and a "U-shaped" relationship between environmental regulation and quantitative economic development. Strict environmental regulations can inhibit economic growth.

Both the above research and this paper believe that environmental regulation can inhibit economic growth. However, the difference is that this paper believes that the inhibition effect of MI and PT environmental regulation on economic growth will be weakened after reaching a certain critical point, while the above research believes that environmental regulation will promote the economy after reaching a certain critical point. Such differences may be caused by the regional heterogeneity of environmental regulation. Tian conducted a study based on the data of provinces and cities across the country, while Ma considered the regional heterogeneity of environmental regulation and divided China into eastern, western and central regions for research. However, provinces and cities in the Yangtze River Economic Belt are mainly located in the central and southern parts of China. The main purpose of this paper is to make up for the deficiency of the relationship between environmental regulation and economic growth in the Yangtze River Economic Belt, an important economic development region.

## 5.2. The impact of market incentivization on regional economic growth

We verified that the second hypothesis is true; that is, the implementation of MI environmental regulation has an inhibitory effect on economic growth, and the inhibitory effect is significant in the early stage but weakens when it reaches a certain critical point.

The government has adopted MI environmental regulation, set up enterprise environmental performance and established an incentive system that combines the best tax on industrial emissions with the emission subsidy (polluter payment/emission subsidy) [47]. This system provides financial compensation for enterprises that comply with the regulations, and gives certain punishments to violators, which can mobilize the independence of enterprises. This model appropriately reduces the requirements at the initial stage of the establishment of environmental performance and tries to reduce the impact of relatively significant inhibition on the economic growth of enterprises. After a certain period of time, it can improve the environmental performance requirements, which can not only help enterprises to smoothly adapt to the impact of the initial stage of the establishment of environmental regulations on the economic development of enterprises, but also encourage enterprises to use the performance bonus to develop, purchase and apply corresponding environmental protection technologies.

The pressure of environmental regulation on enterprises basically comes from reducing environmental pollution and improving enterprise performance. Liang et al. [40] believes that environmental regulation has an inhibiting effect on enterprises' green technology innovation,

because environmental regulation increases enterprise costs and reduces enterprise performance. Adopting MI environmental regulations, encouraging enterprises' research and application of environmental protection technologies, giving certain economic incentives to breakthroughs in technological development, and compensating for the impact of technological research and development costs on enterprises' economic growth can minimize the inhibitory effect on regional economic growth on the premise of ensuring the advantages of MI environmental regulations.

## 5.3. The impact of public environmental regulation on regional economic growth

We verified that the third hypothesis is true; that is, the implementation of PT environmental regulation has an inhibitory effect on economic growth, and the inhibitory effect is significant in the early stage, but decreases after reaching a certain critical point.

PT environmental regulation requires public participation in environmental governance, and public appeal is conducive to improving green total factor productivity (GTFP) [48]. When negotiating investment matters with enterprises, the regional government can take into account the inhibition effect of public environmental demands and environmental regulations on the early stage of economic growth. On the premise that enterprises pay attention to public demands for investment, the regional government will give enterprises greater policy support in the early stage of establishing enterprises locally and reduce the impact of the inhibition effect on the economic growth of enterprises in the early stage. After the development of enterprises becomes stable, some measures need to be taken to give back to the local community. The regional government can thus exchange for richer and more beneficial returns in the later period.

In the era of economic globalization, a large number of foreign investors have poured into the domestic market to seek cooperation and development. The inflow of foreign direct investment (FDI) is affected by many factors, and strict environmental regulations will not inhibit the inflow of FDI. High-quality FDI tends to flow into regions with orderly social environments, potential economic markets and industrial agglomeration [49]. The adoption of PT environmental regulation can bring about regional policies with high compatibility between public life and market economic development, effectively improve the quality of FDI and have a positive effect on regional economic growth.

## 6. Conclusions and policy recommendations

### 6.1. Research findings

Based on the panel data for 11 provinces (municipalities) in China's Yangtze River Economic Belt from 2014 to 2021, we used the panel threshold model to empirically analyze the impact of three types of environmental regulations on economic growth: CAC, MI and public participation. The conclusions are as follows. Using the threshold model, the research assumptions in this paper were tested; H2 and H3 were supported, and H1 was not verified.

### 6.2. Policy recommendations

Firstly, As the conclusion of this paper is that MI and PT environmental regulations have obvious inhibitory effects on economic growth in the early stage, it is suggested that the government should give economic incentives to the development and breakthrough of environmental protection technologies of enterprises in the early stage of implementing these

two environmental regulations, so as to compensate for the impact of technology research and development costs on economic growth of enterprises.

Second, the government should help companies establish and broaden open channels for environmental information. In the early stage of the implementation of environmental regulations, the public is called on to participate in the environmental protection work with simple operation and strong interaction, so as to save some environmental protection funds for enterprises.

Third, market feedback mechanisms need to be strengthened, and the government should adjust its policies immediately. In the implementation of market incentive environmental regulation, the government pays attention to the economic development of enterprises, establishes a sensitive feedback mechanism, and comprehensively uses environmental regulation means such as pollution charges, environmental protection subsidies, and pollution supervision to adjust relevant fees in a timely manner. These measures help enterprises to ease the economic pressure in the early stage of the implementation of environmental regulations.

## 6.3. Research prospects

Protecting the environment and economic growth is a difficult problem that cannot be avoided in the context of contemporary China's development. It is also the key to establishing an environmentally friendly society. In this article, we compared the provinces and cities in the Yangtze River Economic Belt region as the research object, and further examined the impact and path of environmental regulation based on CAC, MI and public participation, which has certain reference significance for promoting China's economic growth. In terms of the classification of environmental regulations, there are many existing classification methods. In addition to the CAC, MI and public participation environmental regulations mentioned in this article, there are various other classifications such as investment, cost and voluntary. In terms of the availability and authenticity of data, we only selected some classifications for analysis. The next step in the research should consider a reasonable and comprehensive definition of the classification of environmental regulations and search for relevant data for empirical analysis.

## Author Contributions

**Conceptualization:** Jiajun He, Hui Zhang.

**Data curation:** Mingwei Song.

**Funding acquisition:** Jiajun He, Mingwei Song.

**Methodology:** Xin Fan.

**Project administration:** Hui Zhang.

**Software:** Zirui Huang.

**Supervision:** Hui Zhang.

**Validation:** Zirui Huang.

**Writing – original draft:** Jiajun He, Rong Zhou.

**Writing – review & editing:** Zirui Huang, Hui Zhang, Mingwei Song.

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
