## [Decision Letter · Decision Letter 0]

4 Jul 2023

PONE-D-23-16688The Impact of Environmental Regulation on Regional Economic Growth: a Case Study of the Yangtze River Economic Belt, ChinaPLOS ONE

Dear Dr. Zhang,

Thank you for submitting your manuscript to PLOS ONE. After careful consideration, we feel that it has merit but does not fully meet PLOS ONE’s publication criteria as it currently stands. Therefore, we invite you to submit a revised version of the manuscript that addresses the points raised during the review process.

We look forward to receiving your revised manuscript.

Kind regards,

Najabat Ali, Ph.D.

Academic Editor

PLOS ONE

Journal Requirements:

   "Thanks to financial support from the Fundamental Research Funds for the Central Universities and the National Natural Science Foundation of China."

   "This research was funded by the Fundamental Research Funds for the Central Universities (2662022ZHYJ004) and the National Natural Science Foundation of China (40971054)."

   "This research was funded by the Fundamental Research Funds for the Central Universities (2662022ZHYJ004) and the National Natural Science Foundation of China (40971054)." 

6. Please amend the manuscript submission data (via Edit Submission) to include author Mingwei Song.

7. Please include your tables as part of your main manuscript and remove the individual files. Please note that supplementary tables (should remain/ be uploaded) as separate "supporting information" files

Reviewers' comments:

Reviewer's Responses to Questions

**Comments to the Author**

1. Is the manuscript technically sound, and do the data support the conclusions?

Reviewer #1: Partly

Reviewer #2: Yes

2. Has the statistical analysis been performed appropriately and rigorously? 

Reviewer #1: Yes

Reviewer #2: Yes

3. Have the authors made all data underlying the findings in their manuscript fully available?

Reviewer #1: No

Reviewer #2: Yes

4. Is the manuscript presented in an intelligible fashion and written in standard English?

Reviewer #1: Yes

Reviewer #2: Yes

5. Review Comments to the Author

Reviewer #1: PONE-D-23-16688

Title: The Impact of Environmental Regulation on Regional Economic Growth: a Case Study of the Yangtze River Economic Belt, China.

Comments:

I have completed the review process of the article entitled " The Impact of Environmental Regulation on Regional Economic Growth: a Case Study of the Yangtze River Economic Belt, China.". The following comments might further improve the quality of the paper.

1. In the introduction, the theoretical and technical information between is not given. The subject is covered with general statistics and rankings. If variables are added to the model for analysis, their theoretical and technical connections and background should be explained. Otherwise, every desired variable should not be added to the model. In addition, the channels through which these explanatory variables affect the dependent variable should be explained.

2. The gap in the literature should be explained in detail and the importance and contribution of the study should be clearly stated.

The literature review section is thin with respect to the major study areas, it is required substantial improvement.

4. There are many papers cited in the methodology with any logical reason. Add some advantages of the methods, by citing them with reference to the previous studies that make sense.

3. While presenting the empirical results, the study should focus to compare the results obtained and the results of previous studies, and discuss the differences discovered. Are there any other studies in the literature that found similar results? Do the findings overlap with only one study?

4. Conclusion restates the findings, which is inappropriate. It is recommended that authors use no more than three sentences to sum up the findings. More importantly, the policies are extremely ambiguous and appear to have been predetermined by the authors before the study was even started. The policy recommendations should follow logically from the presentation of the findings and not add anything new.

5. I also think that the study needs proof and editing. I would recommend checking it especially in terms of typo errors.

Reviewer #2: I congratulate the authors for a significant contribution in the field and therefore I recommend the manuscript for publication in its current form. I hope, the authors will continue their efforts to make contribution in their field in future.

6. PLOS authors have the option to publish the peer review history of their article (what does this mean?). If published, this will include your full peer review and any attached files.

Reviewer #1: No

Reviewer #2: No

---

## [Author Response · Author response to Decision Letter 0]

21 Jul 2023

Journal Requirements

1 Please ensure that your manuscript meets PLOS ONE's style requirements, including those for file naming.

Reply: We have modified the manuscript as the journal’s requirements. 

2 Thank you for stating the following in the Acknowledgments Section of your manuscript: "Thanks to financial support from the Fundamental Research Funds for the Central Universities and the National Natural Science Foundation of China."

Please remove any funding-related text from the manuscript and let us know how you would like to update your Funding Statement. Currently, your Funding Statement reads as follows: "This research was funded by the Fundamental Research Funds for the Central Universities (2662022ZHYJ004) and the National Natural Science Foundation of China (40971054)."

Reply: Thank you for your kindly reminder. We have removed the acknowledgments and the Funding. 

The Funding Statement reads as follows: "This research was funded by the Fundamental Research Funds for the Central Universities (2662022ZHYJ004) and the National Natural Science Foundation of China (40971054)."

3. Thank you for stating the following financial disclosure: "This research was funded by the Fundamental Research Funds for the Central Universities (2662022ZHYJ004) and the National Natural Science Foundation of China (40971054)." 

Reply: "The funders had no role in study design, data collection and analysis, decision to publish, or preparation of the manuscript."

Reply: Data Availability statement has been revised. The DOIs is: 10.6084/m9.figshare.23702052 

5.PLOS requires an ORCID iD for the corresponding author in Editorial Manager on papers submitted after December 6th, 2016. Please ensure that you have an ORCID iD and that it is validated in Editorial Manager. To do this, go to ‘Update my Information’ (in the upper left-hand corner of the main menu), and click on the Fetch/Validate link next to the ORCID field. This will take you to the ORCID site and allow you to create a new iD or authenticate a pre-existing iD in Editorial Manager. Please see the following video for instructions on linking an ORCID iD to your Editorial Manager account: https://www.youtube.com/watch?v=_xcclfuvtxQ

Reply: Thank you for your kindly reminder. The ORCID iD has been updated. It is: https://orcid.org/0000-0002-9633-0201

6. Please amend the manuscript submission data (via Edit Submission) to include author Mingwei Song.

Reply: Thank you for your kindly reminder. The manuscript submission data has been revised.

7. Please include your tables as part of your main manuscript and remove the individual files. Please note that supplementary tables (should remain/ be uploaded) as separate "supporting information" files

Reply: Thank you for your kindly reminder. The manuscript submission data has been revised.

Reviewers’ comments

Reviewer #1: 

1.In the introduction, the theoretical and technical information between is not given. The subject is covered with general statistics and rankings. If variables are added to the model for analysis, their theoretical and technical connections and background should be explained. Otherwise, every desired variable should not be added to the model. In addition, the channels through which these explanatory variables affect the dependent variable should be explained.

Reply：Thank you for your comment. We have added theoretical and technical information about both in the introduction. In Part 3.2, we explain the theoretical and technical connections and backgrounds of the variables, and explain the channels through which these explanatory variables affect the dependent variables

2. The gap in the literature should be explained in detail and the importance and contribution of the study should be clearly stated.The literature review section is thin with respect to the major study areas, it is required substantial improvement.

Reply：Thank you for your comment. We have made a large number of revisions to the literature review, deleted the literature with repetitive content, not distinctive features or not closely related to the theme, and explained the characteristics, differences and contributions of the literature. The importance of this research is also pointed out

3. While presenting the empirical results, the study should focus to compare the results obtained and the results of previous studies, and discuss the differences discovered. Are there any other studies in the literature that found similar results? Do the findings overlap with only one study?

Reply：Thank you for your comment. We compared the results of this study with other studies that came to similar conclusions. The differences and their causes are pointed out

4. Conclusion restates the findings, which is inappropriate. It is recommended that authors use no more than three sentences to sum up the findings. More importantly, the policies are extremely ambiguous and appear to have been predetermined by the authors before the study was even started. The policy recommendations should follow logically from the presentation of the findings and not add anything new.

Reply：Thank you for your comment. We cut out the unnecessary content of the research conclusion, and put forward the suggestion that is closely related to the conclusion of this study

5. I also think that the study needs proof and editing. I would recommend checking it especially in terms of typo errors.

Reply：Thank you for your comment. We checked the article and corrected the words and punctuation in question firstly

Reviewer #2:

I congratulate the authors for a significant contribution in the field and therefore I recommend the manuscript for publication in its current form. I hope, the authors will continue their efforts to make contribution in their field in future.

Reply：Thanks very much for your kindly comment.

---

## [Editor Report · Decision Letter 1]

14 Aug 2023

The Impact of Environmental Regulation on Regional Economic Growth: a Case Study of the Yangtze River Economic Belt, China

PONE-D-23-16688R1

Dear Dr. Zhang,

We’re pleased to inform you that your manuscript has been judged scientifically suitable for publication and will be formally accepted for publication once it meets all outstanding technical requirements.

Kind regards,

Najabat Ali, Ph.D.

Academic Editor

PLOS ONE
---

## [Editor Report · Acceptance letter]

30 Aug 2023

PONE-D-23-16688R1 

The Impact of Environmental Regulation on Regional Economic Growth: a Case Study of the Yangtze River Economic Belt, China 

Dear Dr. Zhang:

I'm pleased to inform you that your manuscript has been deemed suitable for publication in PLOS ONE. Congratulations! Your manuscript is now with our production department. 

Kind regards, 

on behalf of

Dr. Najabat Ali 

Academic Editor

PLOS ONE